# Association of MACC1 expression with lymphatic metastasis in colorectal cancer: A nested case-control study

Zheying Zhang[ORCID]¹, Huijie Jia¹, Yuhang Wang¹, Baoshun Du², Jiateng Zhong¹*

**1** Department of Pathology, Xinxiang Medical University, Xinxiang, 453003, P.R. China, **2** Second Department of Neurosurgery, Xinxiang Central Hospital, Xinxiang, 453003, P.R. China

* jtzhong@xxmu.edu.cn

**Data Availability Statement:** All relevant data are within the paper and its Supporting Information files.

**Funding:** This work is supported by National Natural Science Foundation of China (Grant

## Abstract

MACC1 gene is a newly discovered gene and plays an important role in the metastasis of colorectal cancer (CRC). The objective of this study was to investigate whether MACC1 is an independent factor associated with lymphatic metastasis in CRC patients. We analyzed the association between MACC1 expression and lymphatic metastasis in a nested case-control study including 99 cases and 198 matched controls in CRC patients, assessed from August 2001 to March 2015. Cases were defined as lymphatic metastasis and non-lymphatic metastasis according to AJCC TNM stages; for each case, two age-matched control without lymphatic and distant metastasis was randomly selected from the study participants. Demographic, variables about metastasis and MACC1 expression were collected. In multivariate analysis, the OR (95% CI) of MACC1 expression was 1.5 (1.1 to 2.0) in patients with lymphatic metastasis versus non-lymphatic metastasis after adjusting all variables. After adjustment for all variables and age stratification, MACC1 expression was found to be an independent risk factor for lymph node metastasis in the middle-aged group (OR 2.1, 95% CI 1.1–4.0). A nonlinear relationship between MACC1 expression and 64–75 age group was observed. The probability of metastasis slightly increased with the MACC1 level lower than turning point 1.4. At the same time, the probability of lymphatic metastasis was obviously increased even after adjusting all variables when MACC1 level higher than 1.4 (OR 11.2, 95% CI 1.5–81.5; p = 0.017) in the middle age group. The expression of MACC1 was not associated with lymphatic metastasis in populations younger than 64 or older than 75. The results demonstrates that increased MACC1 level in 64–75 age group might be associated with lymphatic metastasis in CRC patients.

## Introduction

Colorectal cancer (CRC) is a common malignant tumor that seriously threatens human life and health. In recent years, with the improvement of people's living standards, especially the change of dietary structure, the incidence and mortality of CRC have increased year by year [1]. Metastasis is an important cause of death in patients with CRC. More than half of the

No.81802470,U1804173). The Doctoral Scientific Research Foundation of Xinxiang Medical University (Grant No. XYBSKYZZ201632), the Department of Science and Technology of Henan Province (Grant No. 192102310362) and Medical Science and Technology of Henan Province (Grant No. LHGJ20190452). The funders had no role in study design, data collection and analysis, decision to publish, or preparation of the manuscript.

**Competing interests:** The authors have declared that no competing interests exist.

patients with CRC have micro-metastasis before radical surgery, which is the direct cause of recurrence of CRC after surgery. Lymphatic metastasis is the main way of CRC metastasis [2,3]. Therefore, it is important to find the factors that play an important role in the diagnosis and prognosis of CRC metastasis.

With the development of molecular biology, it is possible to predict the metastasis and prognosis of tumors according to their molecular biological characteristics and guide the individualized treatment of tumors. Metastasis-associated in colon cancer1 (MACC1) gene is a newly discovered gene that plays an important role in metastasis of CRC [4]. Since MACC1 was discovered and reported in 2009, it has attracted great interest from researchers all over the world. The MACC1 gene is located on chromosome 7 (7P21.1). It has seven exons and six introns. Its cDNA contains 2559 nucleotide sequences and encodes a binding protein containing 852 amino acids. The domain of MACC1 protein from amino end to carboxyl end is grid protein, EH functional domain, ZU5 domain, SH3 recognition domain and two death domains [5]. MACC1 was first found in human colon cancer tissues [4]. It has been proved that MACC1 is over-expressed in many tumors, but the highest expression level is found in CRC tissues [6]. MACC1 plays an important role in the proliferation and metastasis of many tumors [7]. MACC1 can promote the invasive growth of pancreatic cancer [8] and plays an important role in the proliferation, metabolism, invasion and migration of gastric cancer [9]. In CRC, MACC1 can promote CRC cell proliferation, invasion, metastasis, chemotherapy resistance and maintain stem cell characteristics through various mechanisms [10–12]. There are many reports show that MACC1 is a predictor of the prognosis of colorectal cancer [12–14]. However, no preceding report has considered the independent effects of MACC1 mRNA expression on lymphatic metastasis in CRC patient as far as we know. Lymphatic metastasis eventually leads to hematogenous metastasis, which leads to a poor prognosis. Early prevention and monitoring of lymph node metastasis will improve the survival rate of patients.

In this study, we aimed to evaluate the association between MACC1 expression and lymphatic metastasis in CRC based on TGCA samples and to determine whether MACC1 is an independent predictor of lymphatic metastasis in patients with CRC.

## Materials and methods

### Study population

Colorectal Adenocarcinoma (TCGA, PanCancer Atlas) RNA sequencing data set and corresponding clinical follow-up information were downloaded from the public database cBioPortal (http://www.cbioportal.org/datasets) [15,16]. We carried out a nested case-control study in CRC patients completion date between 14 August 2001 and 31 March 2015. The date of completion date served as the index date. Cases were defined as lymphatic metastasis according to the AJCC Pathology TNM Stage (N1, N1A, N1B, N2, N2A, N2B); for each case, two controls with AJCC Pathology TNM Stage (N0, M0) at the time of the index date of their corresponding cases were selected, matched by age (within two years). In total, the information on 297 cases was extracted. The data and variables were in Supporting Information files.

### Risk factor and other covariates

In the study, the baseline demographic data and medication use were obtained from the download data. We also collected factors associated with metastasis from literature reports. Then, we collected factors associated with MACC1 expression from the co-expression database. The records were cleaned up for analysis by trained research staff.

## Co-expression genes of MACC1

STRING version 11.0 (http://www.string-db.org) and Coexpedia (http://www.coexpedia.org/) are databases of known and predicted co-expression genes [17,18]. In the STRING database, we selected five genes with the highest correlation with MACC1, including RCHY1, MET, EPS15, HGF and CDH1.In the Coexpedia database, we selected the other five genes with the highest correlation with MACC1, including LAD1, KDF1, ITGB8, RAB25 and CLDN7.

## Bioinformatics analysis of MACC1 expression level in CRC

The MACC1 expression level was analyzed in CRC tissues compared with adjacent tissues based on the Oncomine microarray database (https://www.oncomine.org/resource/login.html) and GEPIA database [19,20]. The MACC1 expression level was analyzed in CRC tissues with lymph node metastasis and non-metastatic based on the UALCAN (http://ualcan.path.uab.edu/analysis.html) database [21]. The genes associated with MACC1 were calculated using GEPIA 2 (http://gepia.cancer-pku.cn/index.html) [20].

## Immunohistochemistry (IHC)

CRC samples were collected from The First Affiliated Hospital of Xinxiang Medical College (Xinxiang, China). A total of 70 colorectal cancer samples were collected, including 5 normal controls, 10 colorectal cancer tissues from patients with lymph node metastasis, 10 non-metastatic colorectal cancer tissues, and 10 metastatic lymph node samples in the age group 64–75 and the age group < 64 and > 75, respectively. All patients had written informed consent forms. All cases were pathologically confirmed. Antibody against MACC1 was purchased from AbClone (Cambridge, MA, USA). The tissue microarrays were immunostained for MACC1 as previously described [22]. The MACC1 immunohistochemical score was calculated based on the percentage of positive cells and staining intensity. Positive rates ranged from 0 to 3, with 0 being <10%, 1 being 10–30%, 2 being 31–50%, and 3 being >50%. The staining intensity ranged from 0 to 3, with 0 as no staining, 1 as weak staining, 2 as moderate staining, and 3 as strong staining. The positive rate and staining intensity were scored by double-blind method. The total score for MACC1 expression was calculated as positive rate fraction × stain intensity fraction, with a value range of 0 to 9. MACC1 expression was defined as either "low" (0–4 points) or "high" (5–9 points) [23].

## Statistical analysis

The data distribution of each covariate between lymphatic metastasis and non-lymphatic metastasis groups using $\chi^2$ tests for categorical data and t-test (normal distribution) or Kruskal–Wallis rank sum-test (non-normal distribution) for continuous variables. Univariate logistic regression and multivariate logistic regression models were used to estimate the ORs and 95% CIs to investigate the risk factors associated with lymphatic metastasis. The relationship between MACC1 levels and lymphatic metastasis determined by a smoothing plot, with an adjustment for potential confounders. P value < 0.05 was considered significant. All of the Analyses were performed with the statistical software packages R (http://www.R-project.org) and EmpowerStats (http://www.empowerstats.com).

# Results

## Demographic and baseline characteristics of the study participants

A total of 99 lymphatic metastasis cases and 198 non-lymphatic metastasis controls were included in data analysis in the current study. The mean ages (and standard deviation) of the cases and control subjects were 68.0 ± 12.6 years. The baseline demographic and clinical

characteristics and associated factors for these 297 subjects are summarized in Table 1. Expression levels in the case group of MMP2 were higher in the control group, while the RCHY1 and KDF1 levels were lower compared with the controls. Moreover, the expression level of MACC1 was significantly higher in the case group than in the controls. In addition, tumor recurrence and size were also different between the control group and the case group. Apart from these factors, there was no noticeable difference in the basic characteristics.

## Risk factor and covariates for lymphatic metastasis

The univariate regression analysis showed that MACC1 expression in CRC was significantly correlated with lymphatic metastasis. In addition, new tumor event after initial treatment, pathology tumor size stage, the expression level of RCHY1 and KDF1 might also be associated with lymphatic metastasis (Table 2). In the multivariate logistic regression model for risk factors and covariates associated with lymphatic metastasis, after adjusted for gender and age, the result was the same as univariate regression analysis. But after adjusting for gender, age, the expression level of MET, EPS15, HGF, CDH1, LAD1, ITGB8, RAB25, CLDN7, MMP2, MMP9, VIM, CD44 potential confounding factors, the factors of MACC1 expression, new tumor event after initial treatment and pathology tumor size stage were positively associated

**Table 1. Demographic and clinical characteristics of the cases included in the study (N = 297).**

| Variables | Non-lymphatic metastasis (n = 198) | Lymphatic metastasis (n = 99) | P-value |
|---|---|---|---|
| Age (year) | 68.0 ± 12.6 | 68.0 ± 12.6 | 0.979 |
| Gender | | | 0.804 |
| Male | 113 (57.1%) | 55 (55.6%) | |
| Female | 85 (42.9%) | 44 (44.4%) | |
| New_tumor_event after initial treatment | | | 0.013 |
| No | 139 (70.2%) | 55 (55.6%) | |
| Yes | 26 (13.1%) | 26 (26.3%) | |
| NA | 33 (16.7%) | 18 (18.2%) | |
| Path_T_Stage | | | <0.001 |
| T1, T2 | 57 (28.8%) | 5 (5.1%) | |
| T3, T4 | 141 (71.2%) | 94 (94.9%) | |
| MACC1 | 0.3 ± 1.1 | 0.8 ± 1.3 | 0.002 |
| RCHY1 | 0.1 ± 1.2 | -0.2 ± 0.9 | 0.011 |
| MET | 0.3 ± 1.1 | 0.4 ± 1.3 | 0.435 |
| EPS15 | -0.1 ± 1.0 | -0.2 ± 0.9 | 0.45 |
| HGF | 0.1 ± 1.0 | 0.5 ± 2.6 | 0.061 |
| CDH1 | 0.2 ± 1.1 | 0.3 ± 1.2 | 0.194 |
| LAD1 | 0.1 ± 1.2 | 0.3 ± 1.4 | 0.337 |
| KDF1 | -0.2 ± 1.0 | -0.5 ± 1.1 | 0.021 |
| ITGB8 | 0.1 ± 0.8 | 0.0 ± 1.4 | 0.978 |
| RAB25 | 0.2 ± 1.3 | 0.1 ± 1.0 | 0.624 |
| CLDN7 | -0.4 ± 0.9 | -0.5 ± 1.0 | 0.401 |
| MMP2 | 0.0 ± 0.8 | 0.2 ± 1.2 | 0.047 |
| MMP9 | 0.0 ± 0.8 | -0.2 ± 0.5 | 0.093 |
| VIM | -0.1 ± 0.9 | 0.3 ± 2.7 | 0.078 |
| CD44 | 0.0 ± 1.0 | -0.1 ± 1.1 | 0.204 |

Variables are expressed as n (%) or mean±SD.

**Table 2. Effects of risk factors on lymphatic metastasis by univariate analysis (N = 297).**

| Variables | Total | Odd ratio (95% CI) | P-value |
|---|---|---|---|
| Age | 68.0+12.6 | 1.0 (1.0, 1.0) | 0.979 |
| Gender | | | |
| Male | 168 (56.6%) | 1 | |
| Female | 129 (43.4%) | 1.1 (0.7, 1.7) | 0.804 |
| New tumor event After initial treatment | | | |
| No | 194 (65.3%) | 1 | |
| Yes | 52 (17.5%) | 2.5 (1.4, 4.7) | 0.004 |
| NA | 51 (17.2%) | 1.4 (0.7, 2.7) | 0.336 |
| Path_T_Stage | | | |
| T1, T2 | 62 (20.9%) | 1 | |
| T3, T4 | 235 (79.1%) | 7.6 (2.9, 19.7) | <0.001 |
| MACC1 | 0.5+1.2 | 1.4 (1.1, 1.7) | 0.002 |
| RCHY1 | 0.0+1.1 | 0.7 (0.5, 0.9) | 0.013 |
| MET | 0.3+1.2 | 1.1 (0.9, 1.3) | 0.435 |
| EPS15 | 0.8 | 0.9 (0.7, 1.2) | 0.449 |
| HGF | 0.2+1.7 | 1.2 (1.0, 1.4) | 0.105 |
| CDH1 | 0.2+1.1 | 1.1 (0.9, 1.4) | 0.195 |
| LAD1 | 0.2+1.3 | 1.1 (0.9, 1.3) | 0.336 |
| KDF1 | 0.8 | 0.7 (0.6, 1.0) | 0.022 |
| ITGB8 | 0.0+1.0 | 1.0 (0.8, 1.3) | 0.977 |
| RAB25 | 0.2+1.2 | 1.0 (0.8, 1.2) | 0.623 |
| CLDN7 | 0.6 | 0.9 (0.7, 1.2) | 0.401 |
| MMP2 | 0.0+0.9 | 1.3 (1.0, 1.7) | 0.065 |
| MMP9 | 0.7 | 0.7 (0.5, 1.1) | 0.100 |
| VIM | 0.1+1.7 | 1.2 (0.9, 1.5) | 0.170 |

Variables are expressed as n (%) or mean±SD.

with lymphatic metastasis; while the expression level of RCHY1 and KDF1 were negative with lymphatic metastasis (Table 3).

**Table 3. Multivariate logistic regression model for risk factors associated with lymphatic metastasis (N = 297).**

| Exposure | Model I | | Model II | | Model III | |
|---|---|---|---|---|---|---|
| | Odd ratio (95% CI) | P-value | Odd ratio (95% CI) | P-value | Odd ratio (95% CI) | P-value |
| After_Initial_Treatment | | | | | | |
| No | 1 | | 1 | | 1 | |
| Yes | 2.5 (1.4,4.7) | 0.004 | 2.6 (1.4,4.8) | 0.003 | 3.0 (1.5,5.9) | 0.002 |
| NA | 1.4 (0.7,2.7) | 0.336 | 1.4 (0.7,2.7) | 0.354 | 1.4 (0.7,3.0) | 0.328 |
| Path_T_Stage | | | | | | |
| T1, T2 | 1 | | 1 | | 1 | |
| T3, T4 | 7.6 (2.9,19.7) | <0.001 | 7.7 (3.0,19.8) | <0.001 | 7.2 (2.7,19.4) | <0.001 |
| MACC1 | 1.4 (1.1,1.7) | 0.002 | 1.4 (1.1, 1.7) | 0.002 | 1.5 (1.1, 2.0) | 0.004 |
| RCHY1 | 0.7 (0.5, 0.9) | 0.013 | 0.7 (0.5, 0.9) | 0.012 | 0.8 (0.5, 1.1) | 0.098 |
| KDF1 | 0.7 (0.6, 1.0) | 0.022 | 0.7 (0.6, 1.0) | 0.022 | 0.7 (0.5, 1.0) | 0.063 |

Model I adjust for: None.

Model II adjust for: Gender; Age.

Model III adjust for: Gender; Age; MET; EPS15; HGF; CDH1; LAD1; ITGB8; RAB25; CLDN7; MMP2; MMP9; VIM; CD44.

**Table 4. The association between MACC1 and lymphatic metastasis after stratifing age (N = 297).**

| Age | Tertile 1 (<64) | | Tertile 2 (64–75) | | Tertile 3 (>75) | | Total | |
|---|---|---|---|---|---|---|---|---|
| Model | Odd ratio (95% CI) | P-value | Odd ratio (95% CI) | P-value | Odd ratio (95% CI) | P-value | Odd ratio (95% CI) | P-value |
| Model 1 | 1.1 (0.8, 1.5) | 0.685 | 2.2 (1.4, 3.4) | <0.001 | 1.3 (0.9, 1.9) | 0.162 | 1.4 (1.1, 1.7) | 0.002 |
| Model 2 | 1.0 (0.7, 1.6) | 0.840 | 1.8 (1.1, 3.0) | 0.015 | 1.6 (1.0, 2.5) | 0.051 | 1.4 (1.1, 1.8) | 0.005 |
| Model 3 | 0.8 (0.4, 1.8) | 0.615 | 2.1 (1.1, 4.0) | 0.027 | 1.5 (0.8, 2.7) | 0.200 | 1.5 (1.1, 2.0) | 0.010 |

Model I adjust for: None.

Model II adjust for: Gender; Age.

Model III adjust for: Gender; Age; MET; EPS15; HGF; CDH1; LAD1; ITGB8; RAB25; CLDN7; MMP2; MMP9; VIM; CD44.

## Effects of other covariates on the role of MACC1

We further analyze the effect of covariate stratification on MACC1 and found that variate age has a significant effect on MACC1 after stratifying. After trisecting the age, we found that the level of MACC1 expression was only associated with lymph node metastasis in middle group patients aged 64–75 (Table 4).

## Threshold effect analysis

Furthermore, we analyzed the independent effect of MACC1 expression on lymphatic metastasis; after trisecting the MACC1 expression levels, we found that MACC1 was not associated with lymphatic metastasis (Table 5). We speculate that MACC1 may not linearly relate to lymph node metastasis. After adjusted for other variables and stratifying age, a nonlinear relationship between MACC1 expression and lymphatic metastasis in the middle age group was observed (Fig 1). The probability of metastasis slightly increased with the MACC1 level up to the turning point 1.4. While the MACC1 level was more than 1.4 (OR 11.2, 95% CI 1.5–81.5; p = 0.017), the probability of metastasis was obviously increased even after adjusting all variables (Table 6).

## MACC1 mRNA expression level in CRC

MACC1 was upregulated in CRC tissues compared with adjacent tissues based on the 16 microarrays Oncomine dataset. The right graph shows the results of the Hong database. This database included 12 colon specimens, 70 colorectal cancer specimens. The results showed that MACC1 expression was upregulated 3.733-fold in CRC specimens compared with control specimens (Fig 2A). We used GEPIA 2 (http://gepia2.cancer-pku.cn/#index)

**Table 5. Effect of MACC1 expression on lymphatic metastasis (N = 297).**

| Exposure | Model I | | Model II | | Model III | |
|---|---|---|---|---|---|---|
| | Odd ratio (95% CI) | P-value | Odd ratio (95% CI) | P-value | Odd ratio (95% CI) | P-value |
| MACC1 | 1.4 (1.1, 1.7) | 0.002 | 1.4 (1.1, 1.8) | 0.005 | 1.5 (1.1, 2.0) | 0.010 |
| Tertile | | | | | | |
| Tertile 1 | Reference | | Reference | | Reference | |
| Tertile 2 | 1.3 (0.7, 2.4) | 0.433 | 1.0 (0.5, 2.0) | 0.890 | 1.0 (0.5, 2.1) | 0.967 |
| Tertile 3 | 2.1 (1.1, 3.8) | 0.017 | 1.8 (0.9, 3.5) | 0.081 | 1.7 (0.8, 3.8) | 0.198 |

Model I adjust for: None.

Model II adjust for: Gender; Age.

Model III adjust for: Gender; Age; MET; EPS15; HGF; CDH1; LAD1; ITGB8; RAB25; CLDN7; MMP2; MMP9; VIM; CD44.

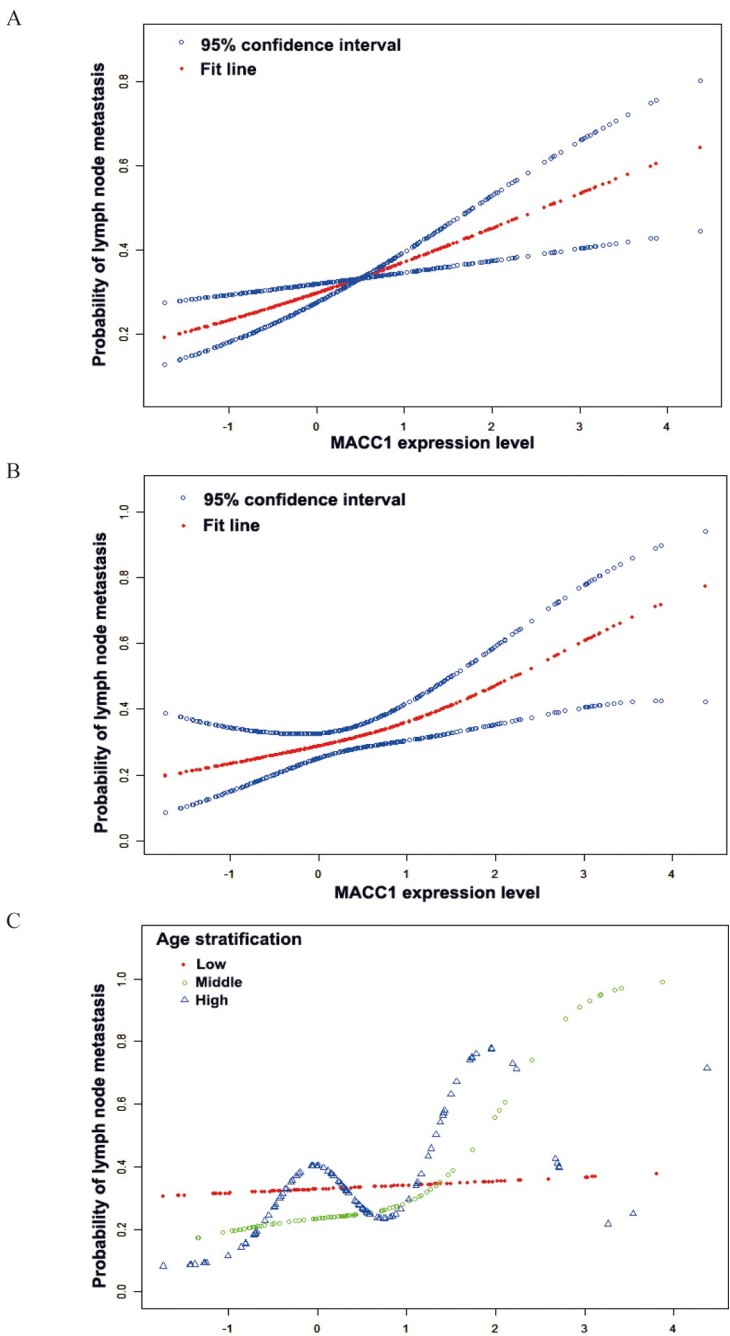

**Fig 1. Relationship between MACC1 expression level and lymphatic metastasis. (A)** A linear relationship between MACC1 expression level and lymphatic metastasis was observed. The blue circle lines on both sides represent 95% confidence interval and the red dot line represents the fitting curves. **(B)** A nonlinear relationship between MACC1 expression level and lymphatic metastasis was observed after adjusting all variables. The blue circle lines on both sides represent 95% confidence interval and the red dot line represents the fitting curves. **(C)** A nonlinear relationship between MACC1 expression level and lymphatic metastasis was observed after adjusting all variables in middle age group. Divide the age into three groups: The red dot line represents the lower age group, the green circle line represents the middle age group, and the blue triangle line represents the higher age group. In this figure, the horizontal ordinate represents the expression level of MACC1, and the longitudinal coordinate represents the probability of lymph node metastasis.

**Table 6. Threshold effect analysis of MACC1 expression on lymphatic metastasis (N = 297).**

| Age | Tertile 1 (<64) | | Tertile 2 (64–75) | | Tertile 3 (>75) | | Total | |
| --- | --- | --- | --- | --- | --- | --- | --- | --- |
| Model | OR (95%CI) | P-value | OR (95% CI) | P-value | OR (95% CI) | P-value | OR (95%CI) | P-value |
| Model 1 | | | | | | | | |
| MACC1 <1.4 | Reference | | Reference | | Reference | | Reference | |
| MACC1 ≥1.4 | 1.5 (0.6,4.1) | 0.392 | 8.0 (2.3,27.9) | 0.001 | 1.7 (0.6,4.4) | 0.305 | 2.4 (1.3, 4.4) | 0.003 |
| Model 2 | | | | | | | | |
| MACC1 <1.4 | Reference | | Reference | | Reference | | Reference | |
| MACC1 ≥1.4 | 1.3 (0.4,4.7) | 0.642 | 7.1 (1.7,29.7) | 0.007 | 3.7 (1.1,12.9) | 0.040 | 3.1 (1.6, 6.1) | 0.001 |
| Model 3 | | | | | | | | |
| MACC1 <1.4 | Reference | | Reference | | Reference | | Reference | |
| MACC1 ≥1.4 | 0.4 (0.1,2.4) | 0.289 | 11.2 (1.5,81.5) | 0.017 | 4.3 (0.9,21.2) | 0.070 | 3.1 (1.4, 6.7) | 0.004 |

OR: Odd ratio.

Model I adjust for: None.

Model II adjust for: Gender; Age.

Model III adjust for: Gender; Age; MET; EPS15; HGF; CDH1; LAD1; ITGB8; RAB25; CLDN7; MMP2; MMP9; VIM; CD44.

online databases in May 2021 to analyze the expression of MACC1 mRNA in Colon adeno-carcinoma (COAD) and Rectum adenocarcinoma (READ) [20]. The results showed that MACC1 upregulated in CRC tissues compared with normal tissues (Fig 2B). We used UAL-CAN (http://ualcan.path.uab.edu/analysis.html) online databases in May 2021 to analyze the relationship between MACC1 expression and lymph node metastasis [21]. The results showed that MACC1 upregulated in CRC tissues with lymph node metastasis compared with non-lymph node metastatic (Fig 2C). The results from GEPIA 2 showed that MACC1 had a relatively high correlation with CDH1 and MET, and a weak correlation with MMP9 and SNAI1. MACC1 is also highly correlated with proliferation-related genes MYC and CDK13 (Fig 3).

## MACC1 protein expression levels in primary foci of CRC patients with and without lymph node metastasis

We examined the expression of MACC1 protein in normal control tissues, colorectal cancer tissues from patients with lymph node metastasis, non-metastatic colorectal cancer tissues, and metastatic lymph node samples patients aged 64–75 years, and patients younger than 64 years and older than 75 years respectively. It was found that MACC1 protein expression was higher in primary foci with lymph node metastasis than in primary foci without lymph node metastasis in the 64–75 age group. It is rarely expressed in normal tissues. The expression was up-regulated in metastatic lymph nodes compared with normal tissues, but not significantly (Fig 4).

## Discussion

The latest global cancer statistics show that the incidence rate of colorectal cancer ranks third, after lung cancer and breast cancer, accounting for about 10.2%. The case fatality rate was the second-highest at 9.2%, second only to lung cancer [24]. About 20% to 25% of the new cases were metastatic colorectal cancer (mCRC) at the time of initial diagnosis. About one-third of the patients receiving radical treatment will eventually relapse into mCRC [25]. It can be seen that mCRC is the main cause of death. Faced with metastasis threatening human health, we

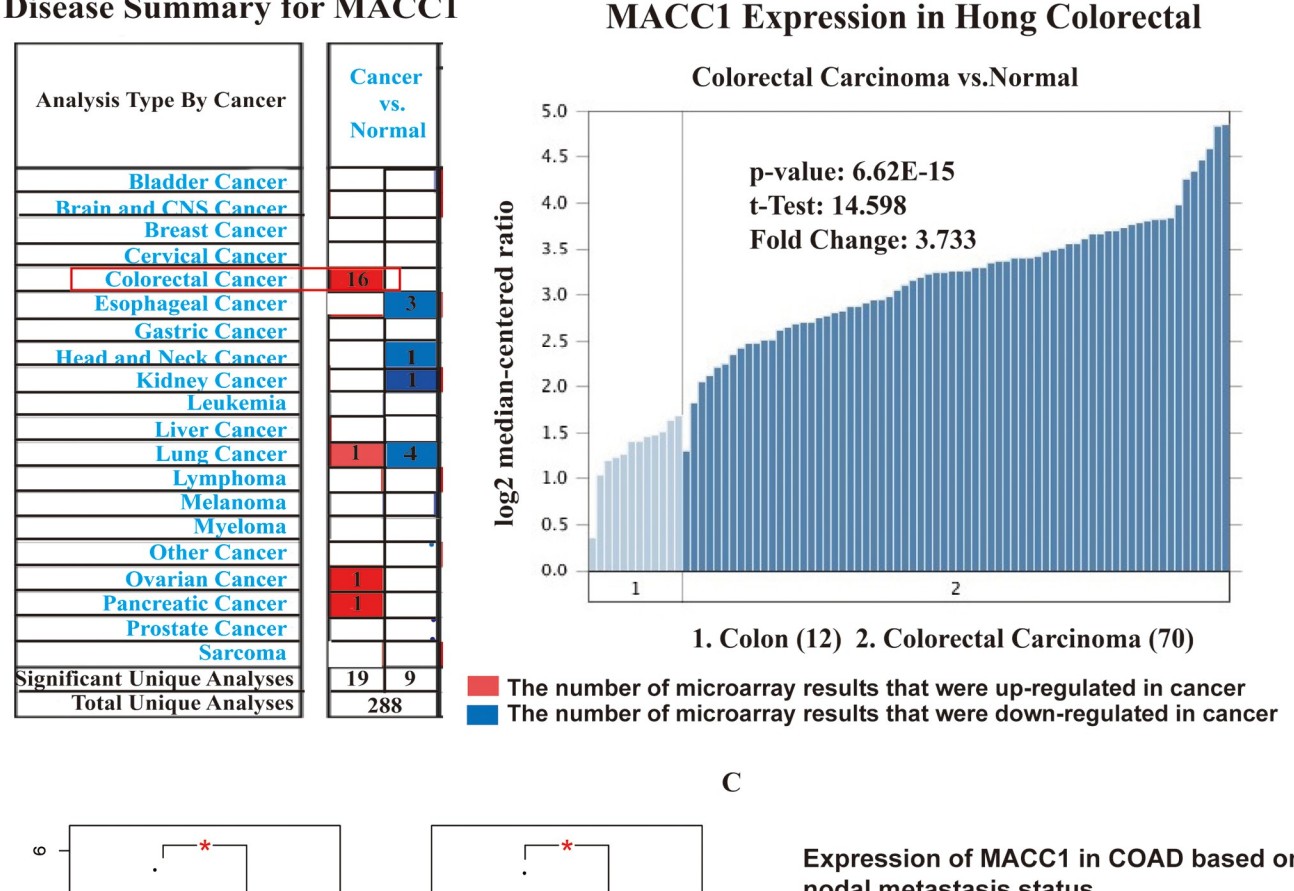

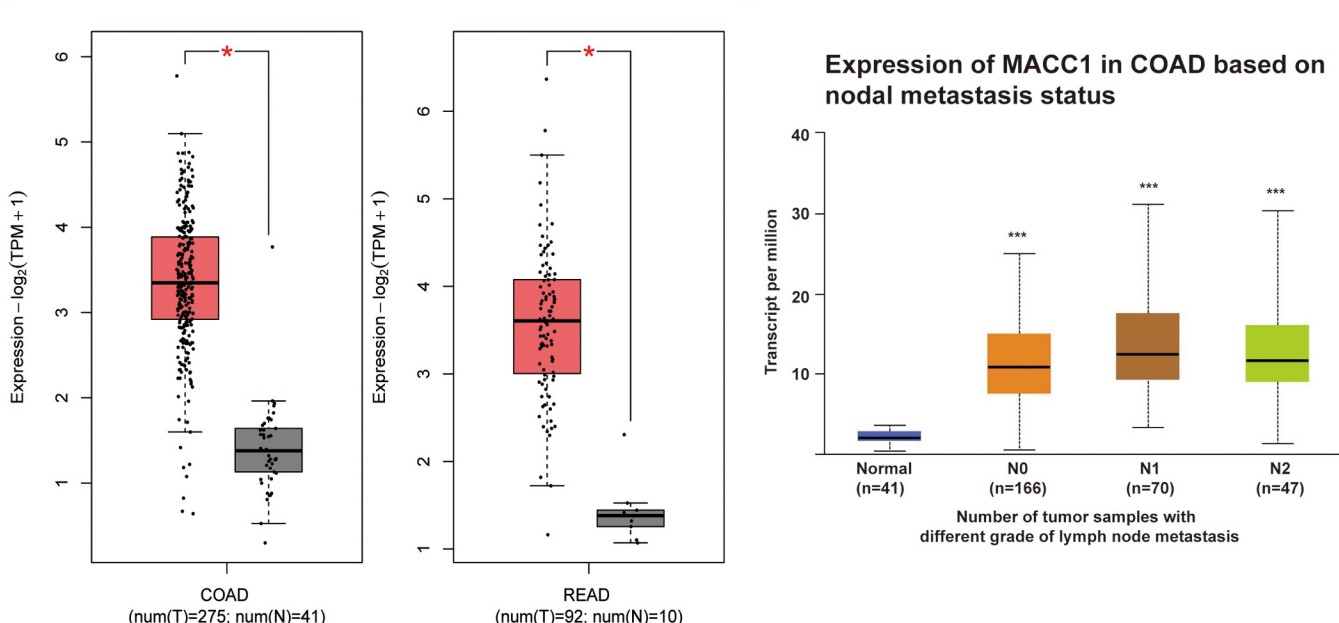

**Fig 2. Expression levels of MACC1 mRNA in CRC tissues.** **(A)** Expression levels of MACC1 mRNA in CRC tissues and normal tissues from Oncomine databse. Red box represents the number of microarray results that were up-regulated in colorectal cancer. Blue box represents the number of microarray results that were down-regulated in colorectal cancer.The graph on the right is the results from Hong Colorectal dataset. The results showed that MACC1 expression was up-regulated 3.733-fold in CRC specimens compared with control colon tissue specimens. **(B)** Expression levels of MACC1 mRNA in CRC tissues and normal tissues from GEPIA 2 database. Red represents the expression in tumor tissue and gray represents the expression in normal tissue. *p < 0.05. **(C)** Expression levels of MACC1 mRNA with lymphatic metastasis CRC tissues and non-lymphatic metastasis control tissues in UALCAN database. The abscissa represents the number of tumor samples with different grade of lymph node metastasis. ***p < 0.001.

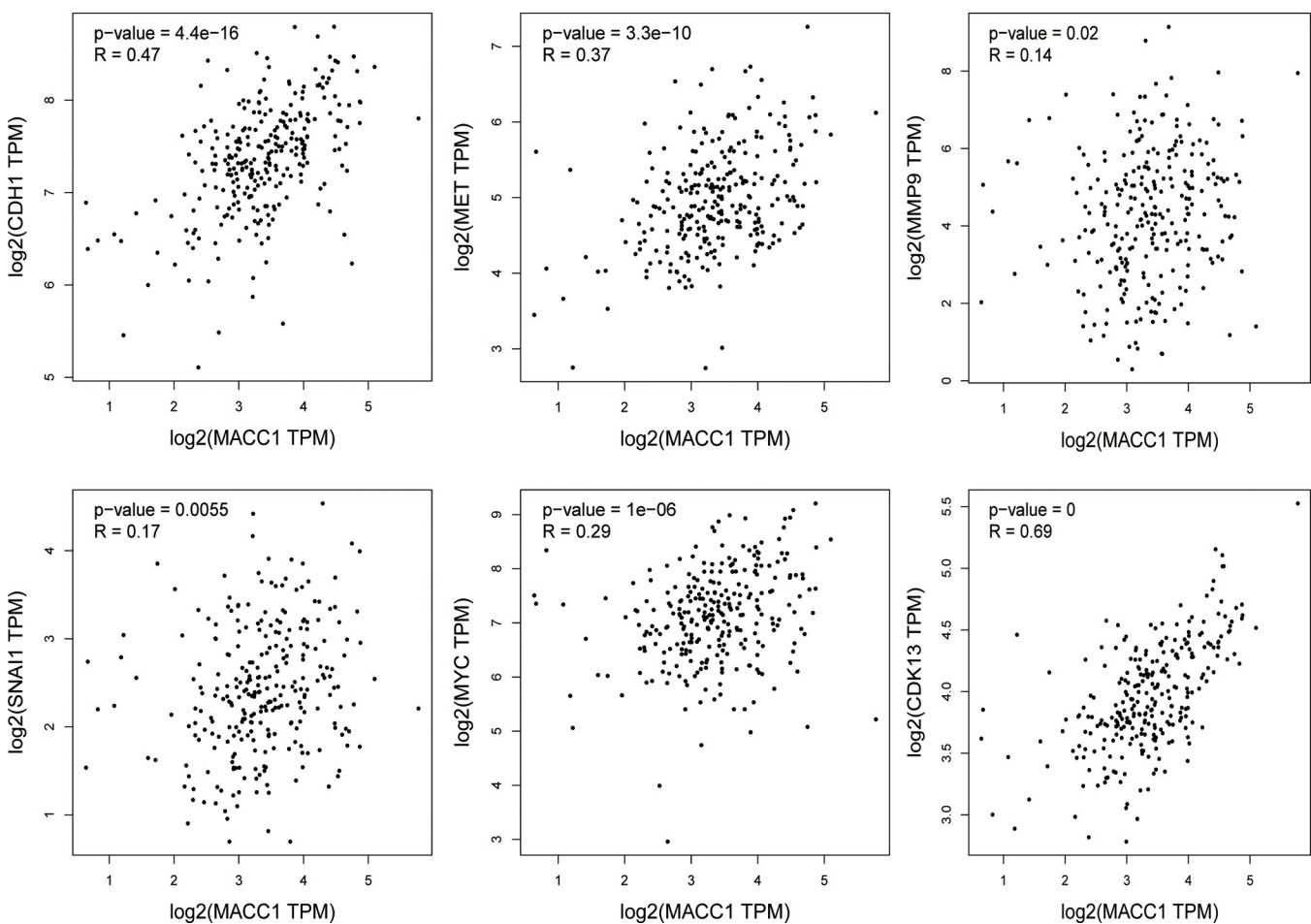

**Fig 3. Bioinformatics analysis of MACC1 co-expression genes.** Pearson's correlation analysis of MACC1 and related genes. We used the services provided by the website GEPIA (http://gepia.cancer-pku.cn/index.html) to examine the correlations between CDH1, MET, MMP9, SNAI1, MYC and CDK13.

should explore effective predictive markers and take early measures to improve prognosis and survival rate.

MACC1 is a new gene discovered by Stein, who analyzed the differential expression in normal colon mucosa, colon adenoma and colon cancer tissues [4]. MACC1 was first found in human CRC tissues. It has been proved that MACC1 is over-expressed in almost all normal human tissues and in many tumors, but the highest expression level is found in CRC tissues [6].

Previous studies have shown that MACC1 play an important role in CRC metastasis, and plasma MACC1 level is an independent prognostic factor for CRC patients [14]. Furthermore, it is reported that MACC1 mRNA level might be a biomarker for poor prognosis in individual Epithelial Ovarian Cancer patients [26]. To date, there are no reports about the independent lymphatic metastasis role of MACC1 in CRC. In this study, we analyzed the relationship between MACC1 expression and lymph node metastasis and the effect of related variables on the relationship. We found both unadjusted and adjusted data showed a significantly positive correlation between MACC1 levels and lymphatic metastasis. Interestingly, when stratifying age factors, We found that MACC1 expression had a correlation with lymph node metastasis only in the middle age group. And a nonlinear relationship was observed between MACC1

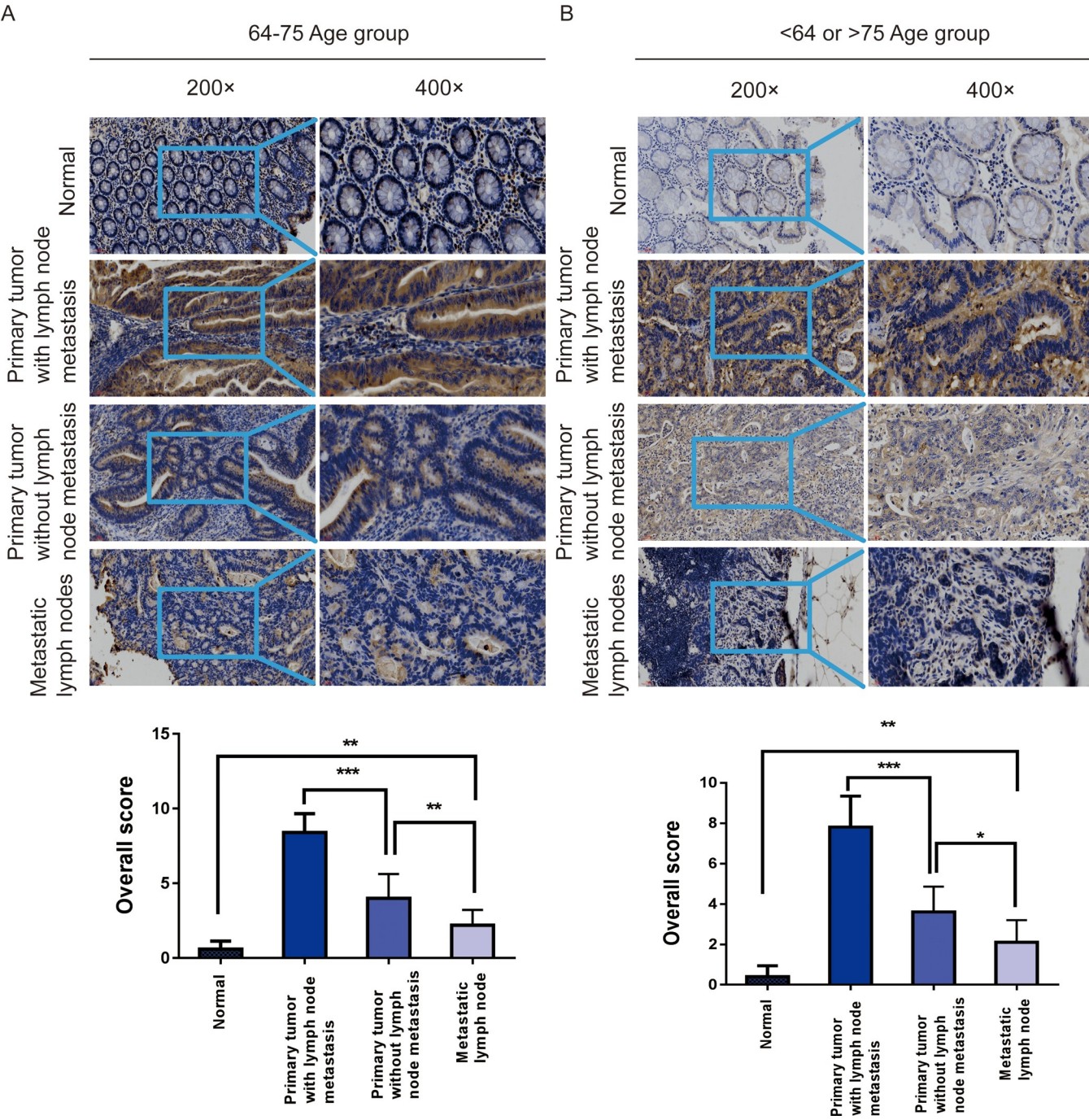

**Fig 4. MACC1 protein expression levels in colorectal cancer tissues with and without lymph node metastasis. (A)** Immunohistochemical method was used to detect the expression level of MACC1 protein in normal control tissues, colorectal cancer tissues from patients with lymph node metastasis, non-metastatic colorectal cancer tissues, and metastatic lymph node tissues in patients aged 64–75 years. **(B)** Immunohistochemical method was used to detect the expression level of MACC1 protein in normal control tissues, colorectal cancer tissues from patients with lymph node metastasis, non-metastatic colorectal cancer tissues, and metastatic lymph node tissues in patients younger than 64 years and older than 75 years. *$p < 0.05$;**$p < 0.01$; ***$p < 0.001$.

expression and lymphatic metastasis in the middle age group. Threshold and saturation effects were then found at the turning point when MACC1 levels reached 1.4. The probability of metastasis slightly increased with the MACC1 level lower than turning point 1.4. While the

probability of metastasis was obviously increased even after adjusting all variables with the MACC1 level higher than 1.4. The level of MACC1 was not associated with lymphatic metastasis in populations younger than 64 or older than 75. This result has a guiding role in the application of clinical MACC1 detection.

We analyzed the expression level of MACC1 mRNA in CRC through an online database. The mRNA level of MACC1 was increased in the cancer tissues and was associated with lymph node metastasis, but there was no difference in the degree of lymph node metastasis. The protein expression of MACC1 was detected by immunohistochemistry. It was consistent with mRNA results. The expression of MACC1 was significantly upregulated in primary lesions in patients with lymph node metastasis compared with patients without lymph node metastasis in the 64–75 age group. These results all indicate that MACC1 plays an important role in CRC lymph node metastasis and can be used as a marker to determine whether lymph node metastasis occurs in the 64–75 age group.

We also analyzed the co-expression of MACC1 and tumor-related genes through the GEPIA database. The results showed that MACC1 had a relatively high correlation with CDH1 and MET, and a weak correlation with MMP9 and SNAI1. MACC1 may play a metastasis role through CDH1 and MET genes, and MACC1 is also highly correlated with proliferation-related genes MYC and CDK13. This suggests that in addition to promoting metastasis, MACC1 may also be closely related to tumor proliferation.

We acknowledged that there are several limitations to the current study. One of the main limitations of this study is the limited sample size, and it would be better if the proportion distribution of the control group to the case group increased. Secondly, not all co-expression factors have yet been described, but too many co-expressed genes that it is not feasible to study all of these factors. Thirdly, We have not yet validated it in different populations, and further studies are needed to validate the results in different cohorts. However, our cases and controls were selected from a well-defined cohort, reducing the likelihood of selection bias and making the difference misclassification of the exposure less likely.

In conclusion, our current study found that MACC1 mRNA expression level has predictive value for the clinical lymphatic metastasis in CRC aged 64–75. These findings may pave the way for the detection of MACC1 in clinical.

## Supporting information

**S1 Data.**
(XLS)

**S2 Data.**
(XLS)

## Author Contributions

**Conceptualization:** Zheying Zhang.

**Data curation:** Huijie Jia, Baoshun Du.

**Formal analysis:** Huijie Jia, Baoshun Du.

**Funding acquisition:** Zheying Zhang, Baoshun Du, Jiateng Zhong.

**Validation:** Yuhang Wang.

**Writing – original draft:** Zheying Zhang.

**Writing – review & editing:** Jiateng Zhong.

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
