## [Decision Letter · Decision Letter 0]

16 Apr 2021

PONE-D-20-41076

Association of MACC1 Expression with Lymphatic Metastasis in Colorectal Cancer: a nested case-control study

PLOS ONE

Dear Dr. Zhong,

Thank you for submitting your manuscript to PLOS ONE. After careful consideration, we feel that it has merit but does not fully meet PLOS ONE’s publication criteria as it currently stands. Therefore, we invite you to submit a revised version of the manuscript that addresses the points raised during the review process.

We look forward to receiving your revised manuscript.

Kind regards,

Ajay Goel

Academic Editor

PLOS ONE

Journal Requirements:

2. Please provide the specific name and/or link to the dataset used in your study, so that readers may easily access the data. In addition, please state the date that you accessed the data.

4.  Please include your tables as part of your main manuscript and remove the individual files. Please note that supplementary tables (should remain/ be uploaded) as separate "supporting information" files

Additional Editor Comments (if provided):

Reviewers' comments:

Reviewer's Responses to Questions

**Comments to the Author**

1. Is the manuscript technically sound, and do the data support the conclusions?

Reviewer #1: Partly

2. Has the statistical analysis been performed appropriately and rigorously? 

Reviewer #1: Yes

3. Have the authors made all data underlying the findings in their manuscript fully available?

Reviewer #1: Yes

4. Is the manuscript presented in an intelligible fashion and written in standard English?

Reviewer #1: Yes

5. Review Comments to the Author

Reviewer #1: This study by Zhang et al. investigated an association of MACC1 with lymphatic metastasis in a nested case control study. Public datasets were employed for bioinformatic analysis and after assessing for MACC1 mRNA levels in different age groups, the authors concluded that an association only applied to a specific age group (64-75 years old). While the study is of interest, due to the lack of clarity with figures presented, it is difficult to assess the validity of their findings.

Comments:

1. Information presented is extensive bioinformatic analysis. Of key significance, missing in this study, is validation at the protein level comparing primary tumors from patients with and without lymphatic metastasis.

2. Authors reach the conclusion that there is a correlation between MACC1 and MMP2, MMP7, CD44, etc). However, according to the Pearson correlation shown of R < 0.5 (Figure 3), this is a very weak association and should be clearly stated in the manuscript.

3. Figure 1 is difficult to interpret. There is no legend to inform the reviewer what each of the lines represent. Figure 1C has a legend in the graph but is blurry and difficult to read.

4. Figure 2A is also blurry and difficult to read.

5. Figure 2C. Unclear what each bar graph represents. Further, what do values on the y-axis represent? Furthermore, does tumor data represent patients with lymphatic metastasis?

6. A more updated reference for #15 should be used. The one cited in manuscript is from 2012.

7. A description of MACC1 (domains, protein size and chromosomal location) do not belong in the discussion session.

8. A few typos and grammatical errors were found.

6. PLOS authors have the option to publish the peer review history of their article (what does this mean?). If published, this will include your full peer review and any attached files.

Reviewer #1: No

---

## [Author Response · Author response to Decision Letter 0]

24 May 2021

Response to Reviewer #1:

Reviewer #1: This study by Zhang et al. investigated an association of MACC1 with lymphatic metastasis in a nested case control study. Public datasets were employed for bioinformatic analysis and after assessing for MACC1 mRNA levels in different age groups, the authors concluded that an association only applied to a specific age group (64-75 years old). While the study is of interest, due to the lack of clarity with figures presented, it is difficult to assess the validity of their findings.

Response: Thank you for your review. We increased the clarity of the image, and marked the vertical and horizontal coordinates in detail, and explained them in detail in the Figure legends. The expression level of MACC1 protein (Figure 4) and mRNA (Figure 2C) in colorectal cancer tissues with and without lymph node metastasis were supplemented. 

Comments:

[1] Information presented is extensive bioinformatic analysis. Of key significance, missing in this study, is validation at the protein level comparing primary tumors from patients with and without lymphatic metastasis. 

Response: Thanks for your kind suggestion. In the results and discussion section, we supplemented the expression level of MACC1 protein in colorectal cancer tissues with and without lymph node metastasis.Thanks!

Supplementary contents in results：

Expression level of MACC1 protein in colorectal cancer tissues with and without lymph node metastasis

We examined the expression of MACC1 protein in normal control tissues, colorectal cancer tissues from patients with lymph node metastasis, non-metastatic colorectal cancer tissues, and metastatic lymph node samples in patients aged 64-75 years, and patients younger than 64 years and older than 75 years, respectively. It was found that MACC1 protein expression was higher in colorectal cancer tissues with lymph node metastasis than in colorectal cancer tissues without lymph node metastasis in 64-75 age group. It is rarely expressed in normal tissues. The expression was up-regulated in metastatic lymph nodes compared with normal tissues, but not significantly (Fig 4).

Figure 4

Fig 4. MACC1 protein expression levels in colorectal cancer tissues with and without lymph node metastasis

(A) Immunohistochemical method was used to detect the expression level of MACC1 protein in normal control tissues, colorectal cancer tissues from patients with lymph node metastasis, non-metastatic colorectal cancer tissues, and metastatic lymph node tissues in patients aged 64-75 years. (B) Immunohistochemical method was used to detect the expression level of MACC1 protein in normal control tissues, colorectal cancer tissues from patients with lymph node metastasis, non-metastatic colorectal cancer tissues, and metastatic lymph node tissues in patients younger than 64 years and older than 75 years. *p < 0.05;**p < 0.01; ***p < 0.001.

Supplementary contents in Discussion：We analyzed the expression level of MACC1 mRNA in CRC through online database. The mRNA level of MACC1 was increased in the cancer tissues and was associated with lymph node metastasis, but there was no difference in the degree of lymph node metastasis. The protein expression of MACC1 was detected by immunohistochemistry. It was consistent with mRNA results. The expression of MACC1 was significantly upregulated in primary lesions in patients with lymph node metastasis compared with patients without lymph node metastasis in 64-75 age group. These results all indicate that MACC1 plays an important role in CRC lymph node metastasis and can be used as a marker to determine whether lymph node metastasis occurs in 64-75 age group.

[2] Authors reach the conclusion that there is a correlation between MACC1 and MMP2, MMP7, CD44, etc. However, according to the Pearson correlation shown of R < 0.5 (Figure 3), this is a very weak association and should be clearly stated in the manuscript.

Response: Thanks for your careful comments. We have made a modification in the results and discussion part of the manuscript. 

Modified contents in Results: “The results from GEPIA 2 showed that MACC1 had a relatively high correlation with CDH1 and MET, and a weak correlation with MMP9 and SNAI1. MACC1 is also highly correlated with proliferation-related genes MYC and CDK13 (Fig 3). ”

Modified contents in Discussion: “We also analyzed the co-expression of MACC1 and tumor-related genes through GEPIA database. The results showed that MACC1 had a relatively high correlation with CDH1 and MET, and a weak correlation with MMP9 and SNAI1. MACC1 may play a metastasis role through CDH1 and MET genes, and MACC1 is also highly correlated with proliferation-related genes MYC and CDK13. This suggests that in addition to promoting metastasis, MACC1 may also be closely related to tumor proliferation.”

Figure 3

Fig 3. Bioinformatics analysis of MACC1 co-expression genes.

Pearson’s correlation analysis of MACC1 and related genes. We used the services provided by the website GEPIA (http://gepia.cancer-pku.cn/index.html) to examine the correlations between CDH1, MET, MMP9, SNAI1, MYC and CDK13.

[3] Figure 1 is difficult to interpret. There is no legend to inform the reviewer what each of the lines represent. Figure 1C has a legend in the graph but is blurry and difficult to read.

Response: Thank you for your question. We relabeled the coordinates in Figure 1 and explained them in detail in the Figure legends.

Figure 1

Figure legend：

Fig 1. Relationship between MACC1 expression level and lymphatic metastasis. 

(A) A linear relationship between MACC1 expression level and lymphatic metastasis was observed. The horizontal ordinate represents the expression level of MACC1, and the longitudinal coordinates represents the probability of lymph node metastasis. The blue circles represent 95% confidence intervals and the red dots represent the fitting curves. (B) A nonlinear relationship between MACC1 expression level and lymphatic metastasis was observed after adjusting all variables. The horizontal ordinate represents the expression level of MACC1, and the longitudinal coordinates represents the probability of lymph node metastasis. The blue circles represent 95% confidence intervals and the red dots represent the fitting curves. (C) A nonlinear relationship between MACC1 expression level and lymphatic metastasis was observed after adjusting all variables in middle age group. The horizontal ordinate represents the expression level of MACC1, and the longitudinal coordinates represents the probability of lymph node metastasis. Divide the age into three groups: the red dots represent the lower age group, the green circles represent the middle age group, and the blue triangle represents the higher age group. 

[4] Figure 2A is also blurry and difficult to read.

Response: Thanks for reviewer’s kind suggestion. We add red and blue box notes to Figure 2A and annotated Figure 2A in detail in the Figure legends.

Figure legend：

Fig 2. Expression levels of MACC1 mRNA in CRC tissues. 

(A) Expression levels of MACC1 mRNA with CRC tissues and control tissues in Oncomine. Red box represents the number of microarray results that were up-regulated in colorectal cancer. Blue box represents the number of microarray results that were down-regulated in colorectal cancer.The graph on the right is the results from dataset. The results showed that MACC1 expression was upregulated 3.733-fold in CRC specimens compared with control colon tissue specimens. 

[5] Figure 2C. Unclear what each bar graph represents. Further, what do values on the y-axis represent? Furthermore, does tumor data represent patients with lymphatic metastasis?

Response: Thank you for your question. We mark the ordinate and describe the picture in detail in the annotation. We removed the unimportant original Figure 2B and changed Figure 2C to Figure 2B, and increased the expression level of MACC1 in Colon adenocarcinoma (COAD) and Rectum adenocarcinoma (READ) (Fig 2B) and lymph node metastasis (Fig 2C). Thanks!

Modified contents in results: We used GEPIA 2 (http://gepia2.cancer-pku.cn/#index) online databases in May 2021 to analyze the expression of MACC1 mRNA in Colon adenocarcinoma (COAD) and Rectum adenocarcinoma (READ) [20]. The results showed that MACC1 upregulated in CRC tissues compared with normal tissues (Fig 2B). We used UALCAN (http://ualcan.path.uab.edu/analysis.html) online databases in May 2021 to analyze relationship between MACC1 expression and lymph node metastasis [21]. The results showed that MACC1 upregulated in CRC tissues with lymph node metastasis compared with non-lymph node metastatic (Fig 2C). 

Figure legend：

Fig 2. Expression levels of MACC1 mRNA in CRC tissues. 

(B) Expression levels of MACC1 mRNA in CRC tissues and normal tissues from GEPIA 2 database. Red represents the expression in tumor tissue and gray represents the expression in normal tissue. *p < 0.05. (C) Expression levels of MACC1 mRNA with lymphatic metastasis CRC tissues and non-lymphatic metastasis control tissues in UALCAN database. The abscissa represents the number of tumor samples with different grade of lymph node metastasis. ***p < 0.001. 

Figure 2

[6] A more updated reference for #15 should be used. The one cited in manuscript is from 2012.

Response: Thanks for reviewer’s kind suggestion. We have updated the reference up to date. 

Modified reference: 

24. Bray F, Ferlay J, Soerjomataram I, Siegel R L, Torre L A,Jemal A, Global cancer statistics 2018: GLOBOCAN estimates of incidence and mortality worldwide for 36 cancers in 185 countries. CA Cancer J Clin. 2018; 68(6): 394-424. 10.3322/caac.21492 

Modified contents in discussion: The latest global cancer statistics show that the incidence rate of colorectal cancer ranks the third, after lung cancer and breast cancer, accounting for about 10.2%. The case fatality rate was the second highest at 9.2%, second only to lung cancer [24].

[7] A description of MACC1 (domains, protein size and chromosomal location) do not belong in the discussion session.

Response: Thanks for reviewer’s careful comments. We have put the description of MACC1 in the introduction part.

[8] A few typos and grammatical errors were found.

Response: We are very sorry for our writing. We have corrected spelling mistakes and asked two English teachers to help modify the manuscript. Thank you!

---

## [Decision Letter · Decision Letter 1]

23 Jun 2021

PONE-D-20-41076R1

Association of MACC1 Expression with Lymphatic Metastasis in Colorectal Cancer: a nested case-control study

PLOS ONE

Dear Dr. Zhong,

Thank you for submitting your manuscript to PLOS ONE. After careful consideration, we feel that it has merit but does not fully meet PLOS ONE’s publication criteria as it currently stands. Therefore, we invite you to submit a revised version of the manuscript that addresses the points raised during the review process.

Your manuscript was re-reviewed by one of the original reviewers. While the article is of interest, one of the reviewers have requested a few minor revisions.

Please submit your revised manuscript in next 2-4 weeks. If you will need more time than this to complete your revisions, please reply to this message or contact the journal office at plosone@plos.org. Please include the following items when submitting your revised manuscript:

We look forward to receiving your revised manuscript.

Kind regards,

Ajay Goel

Academic Editor

PLOS ONE

Journal Requirements:

Reviewers' comments:

Reviewer's Responses to Questions

**Comments to the Author**

1. If the authors have adequately addressed your comments raised in a previous round of review and you feel that this manuscript is now acceptable for publication, you may indicate that here to bypass the “Comments to the Author” section, enter your conflict of interest statement in the “Confidential to Editor” section, and submit your "Accept" recommendation.

Reviewer #1: All comments have been addressed

2. Is the manuscript technically sound, and do the data support the conclusions?

Reviewer #1: Yes

3. Has the statistical analysis been performed appropriately and rigorously? 

Reviewer #1: Yes

4. Have the authors made all data underlying the findings in their manuscript fully available?

Reviewer #1: Yes

5. Is the manuscript presented in an intelligible fashion and written in standard English?

Reviewer #1: Yes

6. Review Comments to the Author

Reviewer #1: Additional information and editing of the content provided in figure legend 1 is strongly recommended. There are two blue lines and information for only one is given and this needs clarification. Next, redundancy in the description on what the x and y coordinates are as it is given for each graph. State it at the very end and just once.

7. PLOS authors have the option to publish the peer review history of their article (what does this mean?). If published, this will include your full peer review and any attached files.

Reviewer #1: No

---

## [Author Response · Author response to Decision Letter 1]

25 Jun 2021

Response to Reviewer #1:

Comments:

[1] Additional information and editing of the content provided in figure legend 1 is strongly recommended. There are two blue lines and information for only one is given and this needs clarification. Next, redundancy in the description on what the x and y coordinates are as it is given for each graph. State it at the very end and just once.

Response: Thanks for your kind suggestion. We have modified the figure legend for Figure 1. The explanation of the two blue lines is added, and the redundant part is removed. Thanks!

Modified contents in Figure legend 1：

Fig 1. Relationship between MACC1 expression level and lymphatic metastasis. (A) A linear relationship between MACC1 expression level and lymphatic metastasis was observed. The blue circle lines on both sides represent 95% confidence interval and the red dot line represents the fitting curves. (B) A nonlinear relationship between MACC1 expression level and lymphatic metastasis was observed after adjusting all variables. The blue circle lines on both sides represent 95% confidence interval and the red dot line represents the fitting curves. (C) A nonlinear relationship between MACC1 expression level and lymphatic metastasis was observed after adjusting all variables in middle age group. Divide the age into three groups: the red dot line represents the lower age group, the green circle line represents the middle age group, and the blue triangle line represents the higher age group. In this figure, the horizontal ordinate represents the expression level of MACC1, and the longitudinal coordinate represents the probability of lymph node metastasis.

---

## [Decision Letter · Decision Letter 2]

19 Jul 2021

Association of MACC1 Expression with Lymphatic Metastasis in Colorectal Cancer: a nested case-control study

PONE-D-20-41076R2

Dear Dr. Zhong,

We’re pleased to inform you that your manuscript has been judged scientifically suitable for publication and will be formally accepted for publication once it meets all outstanding technical requirements.

Kind regards,

Ajay Goel

Academic Editor

PLOS ONE

Additional Editor Comments (optional):

Reviewers' comments:

Reviewer's Responses to Questions

**Comments to the Author**

1. If the authors have adequately addressed your comments raised in a previous round of review and you feel that this manuscript is now acceptable for publication, you may indicate that here to bypass the “Comments to the Author” section, enter your conflict of interest statement in the “Confidential to Editor” section, and submit your "Accept" recommendation.

Reviewer #1: All comments have been addressed

2. Is the manuscript technically sound, and do the data support the conclusions?

Reviewer #1: Yes

3. Has the statistical analysis been performed appropriately and rigorously? 

Reviewer #1: Yes

4. Have the authors made all data underlying the findings in their manuscript fully available?

Reviewer #1: Yes

5. Is the manuscript presented in an intelligible fashion and written in standard English?

Reviewer #1: Yes

6. Review Comments to the Author

Reviewer #1: (No Response)

7. PLOS authors have the option to publish the peer review history of their article (what does this mean?). If published, this will include your full peer review and any attached files.

Reviewer #1: No

---

## [Editor Report · Acceptance letter]

26 Jul 2021

PONE-D-20-41076R2 

Association of MACC1 Expression with Lymphatic Metastasis in Colorectal Cancer: a nested case-control study 

Dear Dr. Zhong:

I'm pleased to inform you that your manuscript has been deemed suitable for publication in PLOS ONE. Congratulations! Your manuscript is now with our production department. 

Kind regards, 

on behalf of

Dr. Ajay Goel 

Academic Editor

PLOS ONE